



# Supercell Convective Environments in Spain based on ERA5: Hail and Non-Hail Differences

Carlos Calvo-Sancho[1], Javier Díaz-Fernández[2], Yago Martín[3], Pedro Bolgiani[2], Mariano Sastre[2], Juan Jesús González-Alemán[4], Daniel Santos-Muñoz[5], José Ignacio Farrán[1] and María Luisa Martín[1,6],

[1]Department of Applied Mathematics, Faculty of Computer Engineering, University of Valladolid, Segovia, Spain.
[2]Department of Earth Physics and Astrophysics, Faculty of Physics, Complutense University of Madrid, Madrid, Spain.
[3]Department of Geography, Faculty of History and Philosophy, University Pablo de Olavide, Sevilla, Spain.
[4]State Meteorological Agency (AEMET), Madrid, Spain.
[5]Danmarks Meteorologiske Institut, Copenhaguen, Denmark.
[6]Institute of Interdisciplinary Mathematics (IMI), Complutense University of Madrid, Madrid, Spain.

*Correspondence to*: Carlos Calvo-Sancho (carlos.calvo.sancho@uva.es)

**Abstract.** Severe convective storms, in particular supercells, are occasionally responsible for a large number of losses of property and damages in Spain. This paper aims to study the synoptic configurations and pre-convective environments in a dataset of 262 supercells during 2011-2020 in Spain. The events are grouped into supercells with hail (diameter larger than 5 cm) and without hail and the results are compared. ERA5 reanalysis data are used to study the synoptic configurations and soundings related to the supercell events at the initial and centroid time. Moreover, temperature, convective available potential energy, convective inhibition, lifting condensation level, level of free convection, height of freezing level, wind shear and storm-relative helicity are determined for each event. The results show that supercells are more frequent in the Mediterranean coast during the warm season. There are statistically significant differences between hail and non-hail events in the mentioned thermodynamic and kinematic-related parameters analyzed, such as supercells with hail environments characterized by higher median values of most-unstable convective available potential energy than supercells without hail.

## 1 Introduction

Convective storms and their associated phenomena (lightning, hail, wind or flash-floods) have a great influence on human activities due to the destructive consequences they may have (Martín et al., 2020; Taszarek et al., 2020a; Rodriguez and Bech, 2021). Europe is regularly threatened by severe convective storms (Dahl, 2006), causing considerable economic loss, social impact, and endangering aviation safety (Nisi et al., 2016; Mohr et al., 2017; Antonescu et al., 2017; Kunz et al., 2020; Chernokulsky et al., 2020 Gatzen; et al., 2020). Thus, improving the knowledge on the genesis and life of convective storms is a constant endeavor in the meteorological community.

Thunderstorm cells can be formed either in a discrete and isolated form, or in large and organized systems, e.g., squall-lines. Based on their structure, organization, and size, three different thunderstorm types are defined by the US National Weather Service (NWS, 2019): ordinaries, multicell and supercells. Concerning supercells, Browning (1962) defines them as a



convectives storm, occurring in a significantly vertically-sheared environment, that contains a deep and persistent mesocyclone, representing the most organized, severe and long-lasting form of isolated deep convection phenomena. These systems are linked with hail reports -including hail diameters larger than 5 cm- and EF2 tornadoes or higher (Duda and Gallus, 2010; Quirantes et al., 2014). Supercells are common phenomena in spring and summertime (Brooks et al., 2019), and can be easily detected through ground-based or satellite lightning detection systems (Bedka et al., 2018; Galanaki et al., 2018). However, the local phenomena associated with these systems, such as hail, require observational reports to be confirmed. As many of these events occur in unpopulated areas, the observed weather reports have a spatial bias toward the most populated areas (Groenemeijer et al., 2017; Edwards et al., 2018). In recent years, thunderstorm reports have increased due to the accessibility of the general population to new technologies, especially thanks to smart-phones and social networks. This has allowed an improvement in databases related to convective storms, with increasing availability of the information on these phenomena (Elmore et al., 2014; Krennert et al., 2018; Taszarek et al., 2020b). Nevertheless, a rigorous quality and validity control, through observational data (radar, satellite…), should be applied to these observational reports to be scientifically valid, e.g., European Severe Weather Database (Dotzek et al., 2009).

Due to orography and land-sea interactions, supercells in Europe tend to be smaller, both horizontally and vertically, than those formed in the US, and therefore show reduced rotation and shorter life spans (Quirantes et al., 2014; Taszarek et al., 2020b). In the particular case of Spain, the study of severe convective storms has grown, extending the knowledge about these systems in recent years. Martin et al. (2020) found more than a hundred supercells per year on average in Spain. Weather environments conducive to severe convective storms have been identified in different studies suggesting that synoptic environments (Merino et al., 2013; Mora et al., 2015), mesoscale characteristics (García-Ortega et al., 2012), orography (Romero et al., 1998) and convective parameters (Calvo-Sancho and Martin, 2021) should be considered together in the research of supercells. Castro et al. (1992) explored the role of topography in the formation and evolution of convective storms in the Ebro Valley (Figure 1a), concluding that mountainous terrain affects the supercells' trajectories and velocities. Studies related to severe weather phenomena, such as hail, lightning, or tornadoes, have been also relevant. Salvador et al. (2021) looked into the charge structure favoring cloud-to-ground lightning in a severe convective storm, concluding that a very high upper-positive charge region is related with a strong updraft and higher cloud-ground rates. Regarding hailstorms, Merino et al. (2019) highlight that the main triggers of convection are thermal instability and low-level convergence. López and Sánchez (2009) reveal the large occurrence of these events in northeastern Spain, causing substantial damages for the local economy, especially in crop fields. Nevertheless, tornadoes occurrence and intensity are not as severe as in other regions of the world due to the absence of wet fluxes inland (Rodríguez and Bech, 2018). Gayà (2011) performed a climatology of tornadoes and waterspouts in Spain and Rodríguez and Bech (2018, 2020) surveyed the mesoscale environments wherein tornadoes and waterspouts formed in the Iberian Peninsula. Both studies reveal that wind shear (WS) plays a more important role than Convective Available Potential Energy (CAPE) in synoptic and mesoscale environments in the cold season.

Since severe supercells have caused substantial property damage and economic losses in recent years in Spain, with around 300 casualties due to severe convective storms from 1987 to 2020 (Consorcio de Compensación de Seguros, 2020), this study





aims to provide a better understanding of the supercell synoptic and mesoscale environments. To do this, a set of events are selected from the supercell database of Martin et al. (2020) which are then categorized into two distinct groups, i.e., with hail diameter larger than 5 cm (SP-HAIL) and without hail (SP-NONHAIL). These systems are analyzed through their synoptic and mesoscale environments using ERA5 reanalysis (Hersbach et al., 2020). ERA5 is a state of the art, high-resolution

reanalysis, which has shown successful results in studies related to severe local storm environments over North America (Coffer et al., 2020; Li et al., 2020; Taszarek et al., 2020a, 2020b), severe convective storms across Europe (Taszarek et al., 2020a, 2020b; Calvo-Sancho and Martín 2021), tornadic environments in the Iberian Peninsula (Rodríguez and Bech, 2020) and microbursts (Bolgiani et al., 2020).

This work is organized as follows. The database and methodology are described in Section 2. Section 3 shows the discussion

of the main results related to synoptic and mesoscale environments associated with SP-HAIL and SP-NONHAIL events. Finally, the main conclusions are summarized in Section 4.

## 2 Data and methodology

### 2.1 Datasets

The supercell sample used is selected from the Spanish Supercell Database (Martín et al., 2020) for the 2011-2020 period. This dataset is formed by confirmed and medium-high confidence (detected in radar images but without direct observation; see Figure 4 in Martín et al. 2020) supercell events through reports from volunteers and collaborators. Thanks to volunteers, 20.5% of the medium-high confidence supercells were confirmed by two-dimensional radar images. In total, this dataset is formed by 1758 supercells, from which 262 of them correspond to confirmed supercells and 1495 to medium-high confidence

supercells. It's worth noting that even if the database covers all the Spanish territories, there are no events reported in the Canary Islands. The database defines the supercell spatial life cycle through an ellipse in a Geographical Information System (Figure 1b). Furthermore, the Spanish Supercell Database collects additional information associated with the events, e.g., hail diameter, tornado intensity. In the current study, only the confirmed supercells are selected and grouped according to the presence of hail with diameters greater than 5 cm (SP-HAIL and SP-NONHAIL). It should be noted that, despite the name

given, SP-NONHAIL supercells may contain small hail (less than 5 cm). The initial formation time ($t_0$) and the ellipse centroid time ($t_c$) of each supercell are selected to evaluate differences in two different life phases of the systems. The $t_0$ is selected to characterize the development phase of the cell, while the $t_c$ corresponds to the maturity state. The centroid location is characterized by the geometric center of the ellipse, being $t_c$ the closest time to the middle point of the supercell lifespan (Figure 1b).

The ERA5 reanalysis (Hersbach et al., 2020) is selected to study the synoptic characteristics and the convective variables involved in the supercell development. This is the 5th generation reanalysis created by the European Centre for Medium-Range Weather Forecasts (ECMWF) It is provided with a horizontal grid resolution of 0.25º x 0.25º, 1 hour as temporal resolution and 37 pressure levels for the vertical resolution, from 1000 hPa to 1 hPa. In the current work, the domain is delimited to



60ºN/20ºN x 30ºW/30ºE to study the environments related to the supercells in Spain. To analyze the mesoscale setting, a

vertical profile of temperature, dew point, geopotential height, pressure, and wind components (u, v) are derived from the

ERA5 grid for each supercell event.

**Figure 1: (a) Domain and orography of the study area (m). White star is referred to Maestrazgo area. (b) Example of a supercell**

**ellipse track from Spanish Supercell Database. Basemap source: Spanish Supercell Database Visor (arcg.is/1bPe9n).**



## 2.2 Compositing methodology

Following the methodology of Calvo-Sancho and Martin (2021) and Gensini et al. (2021), supercell soundings for SP-HAIL and SP-NONHAIL events are built for $t_0$ and $t_c$. Each vertical profile is computed from ERA5 using the average of the nearest 9 grid points to the supercell location, covering an area of 0.75º x 0.75º (Figure S1, Supplementary). Thus, we avoid the
variability that would result from choosing a single grid point and cover an area adequate enough to be representative of the mesoscale conditions. A quality control is carried out to remove any sounding related to convective boundary propagation (Brooks et al. 2003, 2007; Gensini et al. 2021). Accordingly, each vertical profile must record a non-zero Most-Unstable Convective Available Potential Energy (MUCAPE) to be included in the study. Once the vertical profiles are obtained, composites for SP-HAIL and SP-NONHAIL are derived at $t_0$ and $t_c$.

Synoptic patterns composites of both events are created to describe and compare the common large-scale features at $t_0$ and $t_c$. The ERA5 atmospheric fields used to compute the composites are: 500 and 300 hPa geopotential height, mean sea level pressure, dew point, wind direction and wind speed at 10 meter above sea level, 700-400 hPa integrated mean of omega vertical velocity, and 0-6 km WS. These atmospheric variables have been used in studies related to spatial patterns of hailstorms (Merino et al., 2013; Melcón et al., 2017), supercells (Gropp and Davenport, 2018) and thunderstorms (Mora et al., 2015).

**2.3 Convective environments methodology**

To characterize the convective environments, the previous 0.75º x 0.75º grid at $t_c$ is used (Figure S1, Supplementary). Several thermodynamic and kinematic variables are calculated for each vertical profile. The selection of these parameters (Table 1) is based on similar studies related to severe convective storms in US and Europe (Rasmunssen et al., 1998; Kaltenböck et al., 2009; Westermayer et al., 2017; Rodríguez and Bech, 2018, 2020; Taszarek et al., 2020; Davenport, 2021). The 2-meter
temperature (T2M) and dew-point (DWPT) are computed. CAPE and Convective Inhibition (CIN) using most-unstable (MU), mixed-layer (ML) and surface-based (SB) parcels are calculated using the virtual temperature correction (Doswell and Rasmussen, 1994). The deep-layer bulk wind shear over 0-6 km (WS06) and the effective bulk wind difference [EBWD; limited to the layer in which CAPE ≥ 100 J kg⁻¹ and CIN ≥ -250 J kg⁻¹; Thompson et al., 2007] are calculated. . Finally, other parameters relevant to SP-HAIL are also included: ML lifting condensation level (MLLCL), ML level of free convection
(MLLFC), height of freezing level (FZH) and height of wet-bulb freezing level (FZH_W).




**Table 1. Parameters used in the mesoscale settings**

| Parameter | Abbreviation | Units |
|---|---|---|
| **Thermodynamic parameters** | | |
| 2-meter temperature | T2M | ℃ |
| 2-meter dew-point temperature | DWPT | ℃ |
| **Parcel parameters** | | |
| Most-unstable convective available potential energy | MUCAPE | $J\,Kg^{-1}$ |
| Surface-based convective available potential energy | SBCAPE | $J\,Kg^{-1}$ |
| Mixed-layer convective available potential energy | MLCAPE | $J\,Kg^{-1}$ |
| Most-unstable convective inhibition | MUCIN | $J\,Kg^{-1}$ |
| Surface-based convective inhibition | SBCIN | $J\,Kg^{-1}$ |
| Mixed-layer convective inhibition | MLCIN | $J\,Kg^{-1}$ |
| Mixed-layer lifting condensation level | MLLCL | m |
| Mixed-layer level of free convection | MLLFC | m |
| Height of freezing level | FZH | m |
| Height of wet-bulb freezing level | FZH_W | m |
| **Kinematic parameters** | | |
| Deep-layer bulk wind shear over 0-6 km | WS06 | $m\,s^{-1}$ |
| Effective bulk wind difference | EBWD | $m\,s^{-1}$ |
| Storm-relative helicity over 0-1 km | SRH01 | $m^2\,s^{-2}$ |
| Storm-relative helicity over 0-3 km | SRH03 | $m^2\,s^{-2}$ |

The application of the non-parametric Mann-Witney test (Mann and Whitney, 1947) is used to establish statistical differences (at $p < 0.05$) between the SP-HAIL and SP-NONHAIL groups for the above-mentioned parameters. Based on this Mann-Whitney Test, an initial analysis between the $t_c$ and $t_0$ differences is carried out (Table 2). The results show that there are almost no differences between $t_0$ and $t_c$ for the same group of supercells (only MLCIN for SP-NONHAIL is significant). Thus, evaluating each group at two different times will not yield any additional information from analyzing only one time step. When

the type of cells is compared at the same time, most of the variables result statistically different, and only MUCIN and WS06 show a different statistical significance between $t_0$ and $t_c$. As a consequence, for the sake of simplicity, we chose to evaluate only $t_c$ for those results where the assessment of both times would be redundant. This is based on the fact that SP-HAIL and SP-NONHAIL are statistically different for WS06 at $t_c$, but not at $t_0$. This variable is much more interesting than MUCIN, as there are other buoyancy terms which can be evaluated while WS06 is a major factor for severe convective environments

(Weisman and Klepm, 1982; Brooks et al., 2003; Taszarek et al., 2017).





**Table 2. p-values of the Mann-Whitney test for all the variables analyzed for SP-HAIL and SP-NONHAIL events at $t_0$ and $t_c$. p-values equal or lower than 0.05 are in bold.**

| | SP-HAIL $t_c$ SP-HAIL $t_0$ | SP-NONHAIL $t_c$ SP-NONHAIL $t_0$ | SP-HAIL $t_c$ SP-NONHAIL $t_c$ | SP-HAIL $t_0$ SP-NONHAIL $t_0$ |
|---|---|---|---|---|
| **MUCAPE** | 0.95 | 0.31 | **0.00** | **0.00** |
| **SBCAPE** | 0.81 | 0.54 | **0.00** | **0.00** |
| **MLCAPE** | 0.84 | 0.45 | **0.00** | **0.00** |
| **T2M** | 0.67 | 0.14 | **0.00** | **0.00** |
| **DWPT** | 0.80 | 0.13 | **0.00** | **0.00** |
| **SBCIN** | 0.85 | 0.21 | **0.00** | **0.00** |
| **MLCIN** | 0.69 | **0.04** | **0.02** | **0.00** |
| **MUCIN** | 0.99 | 0.16 | 0.12 | **0.04** |
| **MLLCL** | 0.54 | 0.06 | 0.43 | 0.38 |
| **MLLFC** | 0.25 | 0.74 | **0.03** | **0.00** |
| **FZH** | 0.59 | 0.97 | **0.00** | **0.00** |
| **FZH_W** | 0.53 | 0.71 | **0.00** | **0.00** |
| **WS06** | 0.72 | 0.81 | **0.04** | 0.13 |
| **EBWD** | 0.74 | 0.18 | **0.00** | **0.00** |
| **SRH01** | 0.76 | 0.68 | 0.78 | 0.77 |
| **SRH03** | 0.40 | 0.29 | 0.18 | 0.36 |

## 3 Results and discussion

The spatial and temporal distribution of supercells for both SP-HAIL and SP-NONHAIL formed in the Spanish mainland are first assessed. The main results relative to large-scale composites, and the thermodynamic and kinematic variables involved in supercell formation in the domain are presented and discussed in the following two subsections.

The spatial distribution of the reported supercell episodes (Figure 2a) shows that most of the events for both SP-HAIL and SP-NONHAIL took place in the eastern half of Spain. The Ebro Valley and the Mediterranean coastal area accumulate 79.9% of the SP-NONHAIL and 88.3% of SP-HAIL. This is consistent with lightning observations in Spain, as the eastern Iberian System area (white star in Figure 1a) has the highest density of lightning flash per year (Mora et al. 2019). This area favors convective initiation and supercell formation due to low-level convergence (northwesterly-southeasterly and south westerly-easterly winds), upper-level forcing for ascent, low-medium level moist coming from the Mediterranean Sea and strong diurnal heating (Mora et al. 2015).

The temporal distribution of supercell events (Figure 2b) matches with the warmest and stronger insolation months (July and August accumulate 53.3% of the SP-NONHAIL and 74.4% of the SP-HAIL storms) in the study area, since deep convection





is a necessary condition to the formation of supercells (Markowski and Richardson, 2010; Miglietta et al., 2017; Taszarek et al., 2019). This is consistent with other studies on convective storms in Europe that assess the higher thunderstorm frequency

in summertime, when the diurnal heating is stronger (Kotroni and Lagouvardos, 2016; Taszarek et al., 2018; Taszarek et al., 2019). The hourly distribution of the supercells (Figure 2c) shows a concentration of the events during the late afternoon (summer Local Time is UTC+2), shortly after the daily insolation maximum in the study area. However, the results also yield a large persistence of the conditions, as many events are reported well into the evening.



**Figure 2: (a) Location of the dataset events (SP-HAIL and SP-NONHAIL) from 2011 to 2020 in Spain. (b) Monthly supercell distribution. (c) Hourly supercell distribution ($t_0$; UTC).**





### 3.1 Large-scale setting synoptic features

Synoptic pattern composites for the most relevant atmospheric variables in SP-HAIL and SP-NONHAIL events are shown in
this subsection to describe and compare the large-scale characteristics associated with the supercells' formation ($t_0$) and mature
($t_c$) phases.

Non-substantial differences between SP-HAIL and SP-NONHAIL are found in the mean sea level pressure (Figure 3).
However, the 500 hPa geopotential height displays a SP-NONHAIL composite with a deeper trough and weaker geopotential
height gradient, in comparison with SP-HAIL. A similar situation is shown by the 300 hPa geopotential height. This
atmospheric configuration promotes weak WS in upper-levels, which could be indicative of weaker convective environments
(Weisman and Klemp, 1982; Brooks et al., 2003; Taszarek et al., 2017). Although there are differences between SP-HAIL and
SP-NONHAIL, both geopotential configurations promote upper-level positive vorticity advection (not shown) and divergence
over Spain, which favor a stronger upper-level forcing (Markowski and Richardson, 2010). Values of 1.1 Pa$^{-1}$ s$^{-3}$ at 700-400
hPa thickness of **Q**-vector divergence are found over eastern Spain. These values indicate forcing for ascent where supercells
could have been originated by strong convection (Figure S2, Supplementary). Thermal lows (1012 hPa) can also be appreciated
in the center of Spain (Figure 3). These lows are typical of the summer months (Tullot, 2000), promoting east wind flows and
ensuring humidity from the Mediterranean Sea in the supercell formation area favoring the initiation of deep convection. Thus,
a more favorable environment for deep-moist convection should be expected for SP-HAIL, as the corresponding composite
shows a deeper thermal low, covering a larger area and accompanied by an enhanced easterly flow.

Mora et al. (2015) studied electrically severe convective storms in the northern plateau of Spain during 2000-2010, finding
that 31% of these thunderstorm episodes were linked to upper-level troughs. These episodes were characterized by strong
baroclinic small waves and deep troughs at 500 hPa, which is a pattern very similar to the one shown in SP-HAIL and SP-
NONHAIL composites in Figure 3, respectively. Therefore, the results are in line with Mora et al. (2015), showing that
supercell episodes in Spain are associated with troughs at upper and medium levels of the troposphere. Overall, it can be seen
that the higher convective activity is located on the eastern of Spain, corresponding to the right side of the troughs, with the
thermal lows at the center of Spain.



Figure 3: 500 hPa geopotential height (coloured; dam), 300 hPa geopotential height (blue contours; dam) and mean sea level pressure (black lines; hPa) composites for (a) SP-NONHAIL at $t_0$, (b) SP-HAIL at $t_0$, (c) SP-NONHAIL at $t_c$ and (d) SP-HAIL at $t_c$.

One of the main features favoring the deep-moist convection is the moist at lower and medium levels (Taszarek et al., 2019). Figure 4 depicts statistically significant differences in the DWPT values between SP-HAIL and SP-NOHAIL in the Mediterranean Sea and the Atlantic Ocean surrounding Spain, being clearly higher for the first case. Over the land, a notable difference in DWPT is seen for SP-HAIL over the Ebro Valley (Figure 4b, d), along with a stronger eastern wind flow. This would be a result from the geopotential and thermal low configuration described above, which induces the advection of humid

air from the Mediterranean. The favorable environment created is then assisted by a triggering of convection west of the Ebro Valley. The high elevations of the Iberian System (> 1400 meters), with a particular mention for the Maestrazgo area (white star in Figure 1a), reduce the role of convective inhibition (Momblona, 2017), which is also met by the convergence of southwestern and eastern surface winds. This initiates the deep convection process that will later develop in the Ebro Valley,





pushed by the south-westerly flow. This process is consistent with the results of the supercell observations for the period 2011-
2020 (Figure 2a).

**Figure 4: 2-m dew point temperature (contours; ºC) and 10 meters wind (arrows; m s⁻¹) composites for (a) SP-NONHAIL at $t_0$, (b) SP-HAIL at $t_0$, (c) SP-NONHAIL at $t_c$ and (d) SP-HAIL at $t_c$. Black points denote statistically significant differences (p-value < 0.05) in dew point temperature.**

The omega vertical velocity composites show statistically significant differences between SP-HAIL and SP-NONHAIL at $t_c$

(Figure 5). The omega maxima for both supercell groups throughout the life cycle of the systems are located in the Ebro valley

axis and the Iberian System Mountains, where supercells are most common (Figure 2a). This maxima omega area (-0.3Pa s⁻¹)

matches with positive **Q**-vector divergence values (1.1 Pa⁻¹ s⁻³) and convergence of **Q**-vectors (Figure S2, Supplementary) in

SP-HAIL events. The maxima omega vertical velocity at 700-400 hPa thickness in SP-HAIL is larger than SP-NONHAIL at

$t_c$ with higher values of maxima omega in SP-NONHAIL at 850-500 hPa (not shown). These higher SP-HAIL omega vertical

velocity along with the low-level wind convergence (Figure 4b) favor the convection initiation (Markowski and Richardson,





2010). Sustained omega vertical velocities (Figure 5d) and winds convergence (Figure 4d) enhance and reinforce deep-moist convection at $t_c$, favoring large hail formation over the Ebro Valley and the Mediterranean area (Gutierrez and Kumjian, 2021). Vertical WS promotes storm organization and its longevity. However, excessive WS can be unfavorable to weak updrafts in

environments of low instability and, furthermore, can be disadvantageous to convection initiation by increasing entrainment (Markowski and Richardson, 2010). Figure 5 shows similar values of WS06, for both SP-HAIL and SP-NONHAIL, being moderate (10-20 m s$^{-1}$) to strong (> 20 m s$^{-1}$). The conjunction of upper-level forcing (Figure 3), low-level convergence (Figure 4) and strong omega vertical velocity (Figure 5) promotes organization, longevity and severity in the convective storm systems.



**Figure 5: 700-400 hPa omega vertical velocity (contours; Pa s$^{-1}$), 0-6 km WS (green lines; m s$^{-1}$) composites for (a) SP-NONHAIL at $t_0$, (b) SP-HAIL at $t_0$, (c) SP-NONHAIL at $t_c$ and (d) SP-HAIL at $t_c$. Black points denote statistically significant differences (p-value < 0.05) in omega vertical velocity.**



### 3.2 Mesoscale settings

As described in the convective environments methodology (Section 2.3), results of the T2M, DWPT, CAPE, CIN, MLLCL,
MLLFC, FZH and WS variables from the ERA5 database are presented in this subsection. These results are shown as violin
plots, where the probability density distributions of each variable can be seen, as well as the differences between SP-HAIL
and SP-NONHAIL events at $t_c$.

Based on the synoptic compositing methodology, schematic SP-HAIL and SP-NONHAIL composite soundings at $t_c$ and $t_0$ are
determined (Figure 6). In order to show the vertical profile of the largest and severe supercells, the 90th percentile (based on
MUCAPE values) of the Spanish Supercell Database is selected. The 90th percentile vertical profile for each supercell
classification reveals interesting features, particularly on the surface, low-levels, and the convective energy. The composite
sounding for SP-HAIL (Figure 6b, d) displays a larger CAPE area and a better buoyancy distribution than for SP-NONHAIL
(Figure 6a, c). This high value of CAPE (1877.1 J kg$^{-1}$) is strongly associated with vertical accelerations (Markowski and
Richardson, 2010), so hail formation would be favored. CAPE values are larger at $t_0$ than at $t_c$, for both categories. The CIN in
SP-HAIL increases from $t_0$ to $t_c$, i.e., the stability is increasing through the supercell life cycle which could favor sustained
systems and reinforce the vertical motions (Gropp and Davenport, 2018). In contrast, the SP-NONHAIL CIN values do not
substantially change (-137.5 J kg$^{-1}$; Figure 6 a, c). LCL (Figure 6, black dot in panels) and LFC do not show significant changes
between SP-HAIL and SP-NONHAIL events. However, the LCL in SP-HAIL exhibits a higher value at $t_c$ in comparison with
the value at $t_0$. According to Mulholland et al. (2021), a higher LCL is related to the width of the deep convective updraft,
resulting in a wider, deeper, and faster vertical velocity, which would be in line with the omega vertical velocity results (Figure
5b, d). Wind barbs reveal a moderate WS06 for both types of supercells. This helps to organize convection as the negative
effect of precipitation and outflow on the updraft is reduced with large WS values above the updraft height (Markowski and
Richardson, 2010). In addition, SP-HAIL low-level WS is higher than SP-NONHAIL, favoring hail growth (Gutierrez and
Kumjian, 2021). Also, the sounding composites show large wind values in upper-levels (< 400 hPa), which may favor wind
divergence at the upper troposphere and deep-moist convection. The evolution from $t_0$ to $t_c$ depicts a reduction in WS for SP-
HAIL, which is mainly denoted in the wind speed and not in the rotation, contrary to the SP-NONHAIL episodes.



**Figure 6: 90th percentile soundings composites for (a) SP-NONHAIL at $t_0$, (b) SP-HAIL at $t_0$, (c) SP-NONHAIL at $t_c$ and (d) SP-HAIL at $t_c$. Black dot indicates the LCL value.**

The distribution for T2M and DWPT show differences between both types of events, statistically significant (Table 2) for both

variables. Different distributions can be seen in Figure 7. The T2M for SP-HAIL depicts a bimodal distribution with model

values of 26 ºC and 31 ºC, approximately, with less variability than the corresponding SP-NONHAIL figure. Moreover, the

T2M distribution shows a larger median value for SP-HAIL (Table 3), exhibiting a larger variability for SP-NONHAIL. The

T2M maximum (minimum) for SP-HAIL is 35.6 ºC (19.3 ºC), showing a very similar value for SP-NONHAIL while the

minimum is significantly lower (11.9 ºC) in SP-NONHAIL. The DWPT amplitudes are similar with greater median values for

SP-HAIL than for SP-NONHAIL (Table 3). However, the 25th percentile for SP-NONHAIL (8.5 ºC) is significantly lower

than SP-HAIL (12.0 ºC). These differences are mainly originated in the low-level wind flows. In the Spanish Mediterranean

area, Balearic Islands and places favorable for maritime fluxes, the main contributor to low-level moisture is advection from

the warm Mediterranean Sea. However, in the Spanish inland the main contributor would be the evapotranspiration of the crop

fields and vegetation (Vicente-Serrano et al. 2014; Tomas-Burguera et al. 2021), contributing considerably less humidity to

the environment.

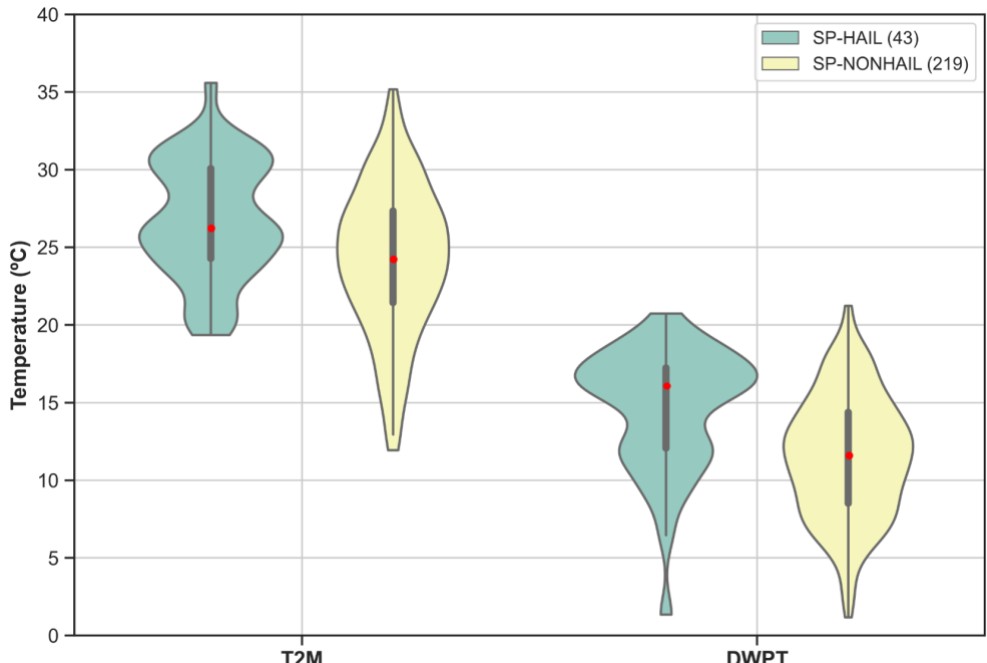

**Figure 7: T2M and DWPT distributions and boxplots for SP-HAIL and SP-NONHAIL at $t_c$. The median is represented as a red point.**

Since CAPE is a common and useful forecast tool for predicting supercells and hail (Kaltenböck et al., 2009; Merino et al.,

2013), its distributions in the parcel measurements of MUCAPE, SBCAPE and MLCAPE are shown (Figure 8a). The results

show statistically significant differences in the distributions for SP-HAIL and SP-NONHAIL (Table 2). It is noteworthy that

CAPE's distributions follow positive skew distributions in SP-NONHAIL events, being the SP-HAIL median values notably

larger than SP-NONHAIL, but with a lower amplitude. However, the 25th percentile and median values for SP-HAIL

SBCAPE, 200 and 600 J kg$^{-1}$, respectively, are lower than those described by Kaltenböck et al. (2009) for Europe,

approximately 400 and 1000 J kg$^{-1}$. Yet, as CAPE is a variable very dependent on humidity and orography, these differences

will arise from the high elevations and relatively low humidity over the domain of study. As expected, MUCAPE and MLCAPE

yield larger values for both SP-HAIL and SP-NONHAIL. Kahraman et al. (2017) analyzed the convective storm environments

for tornado and severe hail days from 1979 to 2013 in Turkey. In their study, severe storm environments are characterized by

smaller CAPE in Turkey compared to US, highlighting that severe hail occurrence is associated with large CAPE and vertical

wind shear. In the current analysis, the median value for MUCAPE and MLCAPE (Table 3) in SP-HAIL events agree with

those obtained in Kahraman et al. (2017) and Púčik et al. (2015) in their study of severe hail-thunderstorms in central Europe.

However, the SBCAPE results are lower than those obtained by Kahraman et al. (2017). This discrepancy might be partially





attributed to the warmer eastern surrounding seas (Mediterranean and Black Sea; Shaltout and Omstedt, 2014). In Taszarek et
al. (2020b) study of severe convective storms with hail, MLCAPE values are greater than for the current study, with values
around 1000 and 1200 J kg$^{-1}$ for the US and Europe, respectively. As it is discussed above, supercells in Europe tend to be
smaller than the ones formed in the US, with lower rotation values and shorter life spans (Quirantes et al., 2014; Taszarek et
al., 2020b).

**Table 3. Median values for each parameter analyzed for SP-HAIL and SP-NONHAIL events at $t_c$.**

|  | SP-HAIL $t_c$ | SP-NONHAIL $t_c$ |
|---|---|---|
| **MUCAPE** | 1031 | 374 |
| **SBCAPE** | 618 | 145 |
| **MLCAPE** | 746 | 203 |
| **T2M** | 26 | 24 |
| **DWPT** | 16 | 12 |
| **SBCIN** | -138 | -77 |
| **MLCIN** | -96 | -54 |
| **MUCIN** | -16 | -16 |
| **MLLCL** | 1661 | 1783 |
| **MLLFC** | 2809 | 2254 |
| **FZH** | 4025 | 3599 |
| **FZH_W** | 3566 | 3238 |
| **WS06** | 19 | 17 |
| **EBWD** | 16 | 10 |
| **SRH01** | 8 | 10 |
| **SRH03** | 88 | 74 |

Related to severe convective storms in Spain, Merino et al. (2013) analyzes several hailstorm days in the Ebro Valley through
high resolution simulations with the WRF model, obtaining a threshold CAPE of 500 J kg$^{-1}$ for hail events. According to
Rodriguez and Bech (2018), the CAPE values found in our study would correspond with those for tornadic storms in eastern
Spain and the Balearic Islands. These authors analyze a dataset of 907 tornado and waterspout events from 1980 to 2018 using
atmospheric profiles from the ERA5 reanalysis and finding SBCAPE values higher than 400 J kg$^{-1}$ in tornadic storms (EF1 or
stronger).

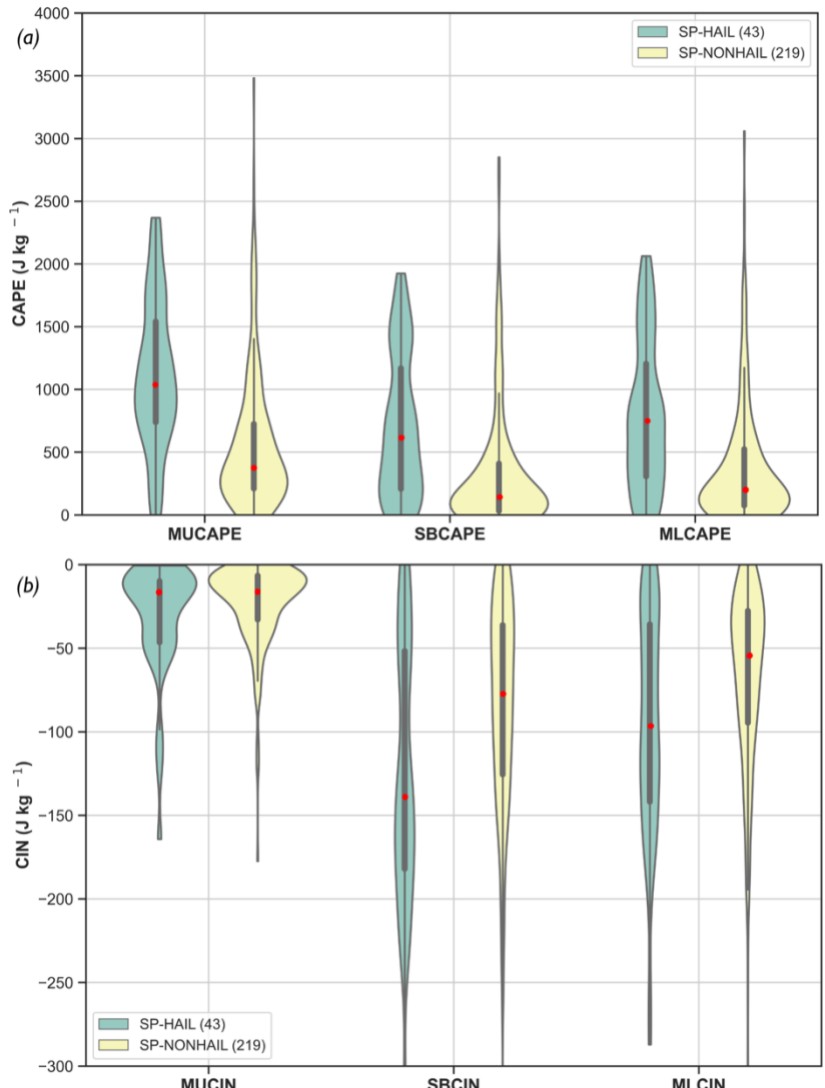

**Figure 8: As in Figure 7, but for (a) MUCAPE, SBCAPE and MLCAPE. (b) MUCIN, SBCIN and MLCIN.**

It is well known that ERA5 CIN values do not represent capping inversions in severe weather environments correctly (Nevius and Evans, 2018; Coffer et al., 2020). Here, CIN distribution in the parcel measures MUCIN, SBCIN and MLCIN are displayed (Figure 8b). The SP-HAIL and SP-NONHAIL differences for SBCIN and MLCIN distributions are statistically significant (Table 2). However, MUCIN differences are not statistically significant, presenting negative skew distributions for both events and absolute values lower than for SBCIN and MLCIN. The SP-HAIL events show an important capping inversion layer (Figure 8b), with a MLCIN median of 95.8 J kg$^{-1}$ (Table 3) and of 53.6 J kg$^{-1}$ for SP-NONHAIL. These results contrast with Taszarek et al. (2020b), who concluded MLCIN values for large hail events are lower (~ 50 J kg$^{-1}$) but similar to those obtained in the US. The higher CIN values obtained in this survey, in comparison with those in Central Europe, could be related with





the average thermal differences in the summer season (3.5 ºC higher than Central Europe; not shown). The strong insolation

and diurnal heating favor this mean temperature difference and, in turn, the thermal low development (Font, 2000). Another cause of the high CIN in Spanish supercells is the complex orography of the Spanish inland. Moreover, enhanced CIN may delay the convective initiation until the CAPE is maximized; once the convection triggers, discrete convective modes, including isolated and elevated supercells, can be developed producing large hail (Rasmussen and Blanchard, 1998; Smith et al., 2012; Thompson et al., 2012, 2013;). Therefore, in light of the SBCIN and MLCIN results, a mechanical trigger to force

the mechanism that initiates convection is required and the conjunction of these factors favors great vertical motions and organized convection.

A comparison of MLLCL, MLLFC, FZH and FZH_W percentiles and distributions between SP-HAIL and SP-NONHAIL is performed in Figure 9. The MLLCL has been an important discriminator between tornadic and non-tornadic supercells in the US (Rasmussen and Blanchard, 1998; Thompson et al., 2003). Taszarek et al. (2020b) compare the MLLCL in severe

convective storms with hail greater than 5 cm in US and Europe obtaining high similarities between both continents. However, in Europe, the MLLCL tends to have much less variability on supercell events, so the skill of this indicator is limited (Kahraman et al., 2017; Taszarek et al., 2020b). The results here shown for Spain are in line with the previous conclusion, since there are no statistically significant differences (Table 2) for MLLCL between SP-HAIL and SP-NONHAIL events. Rodriguez and Bech (2018) also observe this low MLLCL variability between tornadic and non-tornadic convective storms in the Iberian

Peninsula. Nevertheless, Púčik et al. (2015) obtained a MLLCL median value of 1000 m for severe hail-thunderstorms in central Europe, while in the current study the median MLLCL in SP-HAIL events is greater (Table 3). The MLLFC shows significant differences (Table 2) between SP-HAIL and SP-NONHAIL, being higher for SP-HAIL events. It is also remarkable the high variability for both events in comparison with the MLLCL distributions. Comparing the MLLFC values with those obtained by Taszarek et al. (2020) for US and Europe, the domain of study shows a higher altitude and larger variability. The

discrepancy between MLLCL and MLLFC altitudes reflects the high MUCIN values observed in Figure 8. Another important factor to SP-HAIL and SP-NONHAIL events is the freezing level, with differences between both types of supercells statistically significant for FZH and FZH_W (Table 2). Both FZH and FZH_W distributions for SP-NONHAIL events present higher variability than for SP-HAIL ones (Figure 9), being the FZH median value for SP-HAIL higher than for SP-NONHAIL (Table 3).



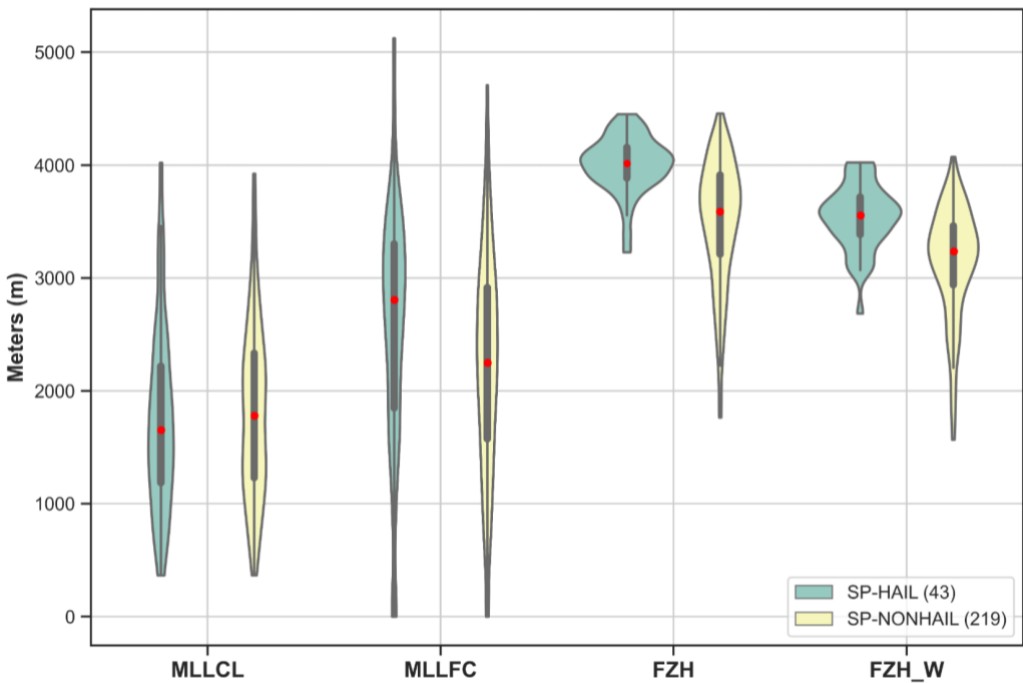


**Figure 9: As in Figure 7, but for MLLCL, MLLFC, FZH and FZH_W.**

Several studies (Rasmussen and Blanchard, 1998; Púčik et al., 2015; Taszarek et al., 2019) suggest that the severity of the convective storms depend on the relationship between CAPE and WS. Furthermore, deep-moist convection tends to develop more organized systems as the WS intensifies (Markowski and Richardson 2010). WS between 0-6 km and EBWD

distributions for SP-HAIL and SP-NONHAIL events are displayed (Figure 10). As expected, the WS06 and EBWD values are higher for SP-HAIL than for SP-NONHAIL with statistically significant differences (Table 2). The WS06 and EBWD median values for SP-HAIL (Table 3) agree with Taszarek et al. (2020b) results for severe convective storms with large hail in Europe; nevertheless, the values for the USA are higher than those presented here. Despite this, SP-HAIL events in the domain of study can be explained by the presence of several mountain ranges in the study area (Figure 1a) where the WS is enhanced by the

interaction of the wind field with orography, with a similar mechanism as observed in the Alps (Kunz et al., 2018; Taszarek et al., 2020). However, ERA5 is limited to reproduce this enhancement due to its horizontal resolution.

Helicity is a frequent parameter used for forecasting supercells and tornadoes since it quantifies the cyclonic updraft rotation in right and left moving supercells (in this survey only the right-moving measure is used; Davies-Jones et al., 1990; Bunkers et al., 2002) Environments with high SRH usually enhance the development of the mesocyclones and large hail formation

(Rasmussen and Blanchard, 1998; Thompson et al., 2003), revealing SRH as a useful predictor for severe weather with organized convection, such as the supercell cases. The SRH01 and SRH03 distributions show no statistically significant differences (Table 2) between SP-HAIL and SP-NONHAIL. The SRH01 variability for both events is quite similar (Figure S3, Supplementary), being the median value of SP-NONHAIL higher than SP-HAIL. On the other hand, it is remarkable the



SRH03 high variability for both events in comparison with the SRH01 distribution. The SP-HAIL mature phase has been
observed in environments with moderate helicity (Table 2), while SP-NONHAIL has lower SRH03 values. There are
remarkable SRH03 maxima for both groups of events with values around 280 m² s⁻² for SP-NONHAIL and 350 m² s⁻² for SP-
HAIL, in line with Kahmaran et al. (2017), who state that large hail occurrences are associated with large values of SRH03.
Rodriguez and Bech (2018, 2021) obtain similar SRH03 values for EF0 and EF1 tornadoes and waterspouts in Iberia. However,
these values are lower than those obtained by Taszarek et al. (2020b) in Europe for large hail reports. The SRH03 results are
also consistent with Calvo-Sancho et al. (2021), where the SRH03 median spatial distribution in Spain displays values of
almost 100 m² s⁻² in the eastern half of Spain.

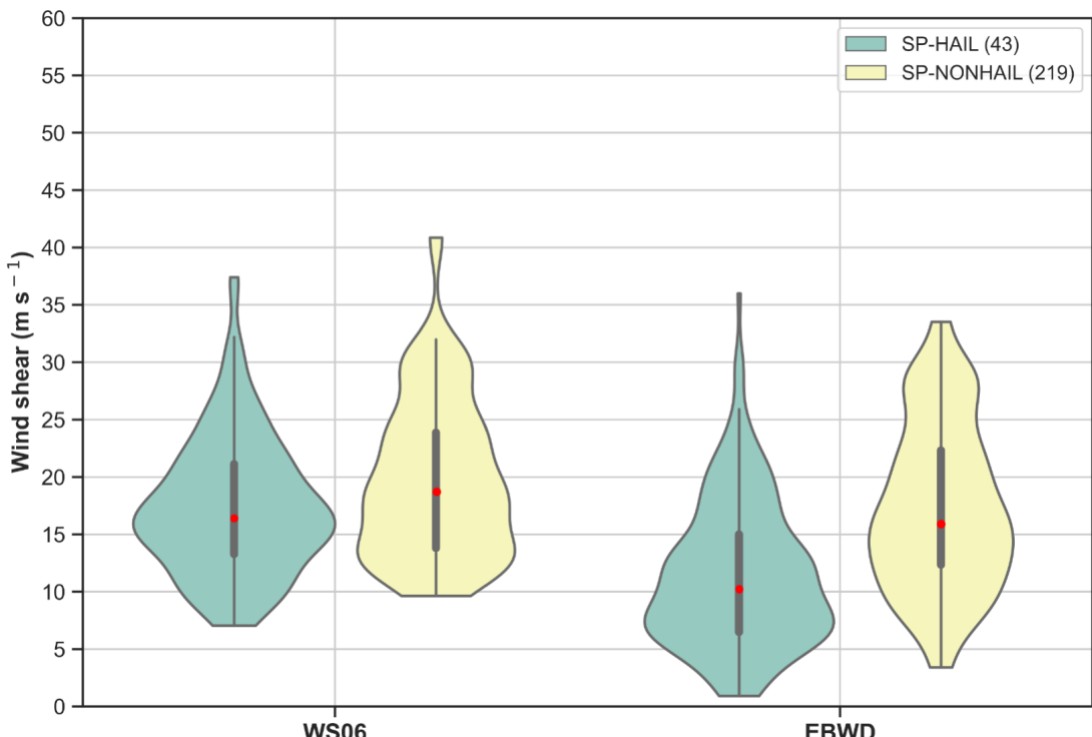

**Figure 10: As in Figure 7, but for WS06 and EBWD.**

## 4 Summary and conclusions

The environments of SP-HAIL and SP-NONHAIL events are here characterized and compared in Spain from 2011 to 2020.
Different atmospheric variables are retrieved from ERA5 reanalysis to obtain the synoptic patterns and sounding composites
at the development time and the mature phase time of the supercells. Thermodynamic and kinematic parameters related to
convective environments are also calculated and compared between SP-HAIL and SP-NONHAIL events at the mature phase
time for the mesoscale environment analysis.





The results yield several conclusions; the most important are listed below:

● There are notable differences in the spatial and monthly distributions of supercells between eastern and western Spain. The eastern half of Spain accumulates 79.9% of the SP-NONHAIL and 88.3% of the SP-HAIL events. July and August accumulate 53.3% of the SP-NONHAIL and 74.4% of the SP-HAIL storms.

● The synoptic patterns composites show a deeper trough for SP-NONHAIL in comparison with SP-HAIL

composites at 500 hPa with the largest height gradients corresponding to SP-HAIL. Strong upper-level forcing is promoted by vorticity advection and upper-level divergence. Surface humidity is influenced by the 10-meters winds, being higher for SP-HAIL. The conjunction of these factors with wind convergences allows the convection initiation. Omega vertical velocity reveals that the SP-HAIL's updraft is higher, allowing the large hail formation.

● The T2M and DWPT values are related to supercell monthly distributions with higher values corresponding to the

warm season and minimum values to the cool season. Both variables are statistically different between SP-HAIL and SP-NONHAIL, being larger for the first group.

● Environments of SP-HAIL events are characterized by approximately three times larger MUCAPE median values than SP-NONHAIL events. Moreover, higher CAPEs, MLLFC, FZH, WS06, EBWD and SRH03 values and lower CIN values are found for SP-HAIL than for SP-NONHAIL. The differences for these parameters between both events are statistically

significant at $p < 0.05$, except for SRH03.

● Based on the ERA5 characterization results for SP-HAIL in Spain, 75% of the events present T2M > 24.3 ℃, DWPT > 12.1 ℃, MUCAPE > 740 J kg$^{-1}$, MLCIN < -36 J kg$^{-1}$, MLLFC > 1850 m, FZH > 3890 m , FZH_W > 3685 m, WS06 > 14 m s$^{-1}$ and SRH03 > 58 m² s$^{-2}$.

Finally, orography and convective environments have been revealed as important factors to supercell formation and

development. Thus, although ERA5 resolution improves previous reanalyses, more research is needed with high-resolution models, allowing the study of the interactions between large-scale and convection processes in the genesis and development of hail supercell events.

**Data availability**

ERA5 reanalysis is available from the Copernicus Climate Change Service Climate Data Store (https://doi.org/10.24381/cds.bd0915c6, Hersbach et al., 2018). Spanish Supercell Database upon request to the author.

**Author contributions**

CC-S, JD-F, YM, MLM, PB and JG-A designed the study. CC-S and JD-F performed the analysis and wrote the first

manuscript. PB, JG-A, MS, and MLM supervised and review. DSM and JF computational and software support. All authors discussed the results and edited the manuscript.



**Conflicts of Interest**

The authors declare that there are no conflicts of interest regarding the publication of this paper.

**Acknowledgements.**

This work was partially supported by the following research projects: PID2019-105306RB-I00 (IBERCANES project), CGL2016-78702-C2-1-R and CGL2016-78702-C2-2-R (SAFEFLIGHT project), FEI-EU-17-16 and SPESMART AND SPESVALE (ECMWF Special Projects). Carlos Calvo-Sancho and Javier Díaz-Fernández acknowledge the grant supported by the Spanish Ministerio de Ciencia, Innovación y Universidades (FPI programs PRE2020-092343 and BES-2017-080025, 420 respectively). Y. Martin acknowledges the grant supported by the European Union (Marie Skłodowska-Curie Programs 101019424).

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
