# Peer review of "Supercell Convective Environments in Spain based on ERA5: Hail and Non-Hail Differences"

_Weather and Climate Dynamics, 2022_

## Author Comment (AC1)

Carlos Calvo Sancho. *e-mail*: carlos.calvo.sancho@uva.es
Department of Applied Mathematics.
Faculty of Computer Engineering. University of Valladolid.
Segovia, Spain.

WEATHER AND CLIMATE DYNAMICS

Manuscript number wcd-2022-27

Segovia, July 12$^{th}$, 2022

Dear Editor:

On behalf of all the co-authors, I am pleased to submit the revised version of the research article entitled "Supercell Convective Environments in Spain based on ERA5: Hail and Non-Hail Differences" by Calvo-Sancho et al. for consideration in the journal Weather and Climate Dynamics.

We thank the reviewers for their time and constructive comments. Before answer all the comments and questions provided by the reviewers it should be considered that changes had been made, following the reviewers' suggestions. In the revised manuscript:

- Only the initial formation time ($t_0$) for each supercell has selected to evaluate the differences between SP-HAIL and SP-NONHAIL events.
- Each vertical profile has computed from ERA5 to characterize the convective variables using the nearest grid point to the supercell location at $t_0$.
- 137 ERA5 reanalysis hybrid model levels have selected as vertical resolution (37 pressure levels were considered in the old manuscript).

Despite the changes abovementioned, the general results of this manuscript don't have significantly change. Please find attached the detailed answers to all the comments and queries provided by the reviewers.

Thank you very much.

Yours sincerely,

Carlos Calvo Sancho

Dear Reviewer 1,

We would like to thank you for your time and effort in our manuscript. We really appreciate the detail of your reading, as your comments have been very useful in improving the paper. We have tried to address all your questions. Please, find below our replies to each one.

Major comments

- Section 3.2. I would not refer to mesoscale settings by using the ERA5 database. I encourage the authors to change the title of this subsection, as well as all mention of mesoscale throughout the paper

Thank you for your suggestion. We agree and we have changed the title of Section 3.2 as *convective variables* and all mentions thorough the paper.

Minor comments

- Line 32: "a convectives storm". Write this sentence in a proper English.

Thank you for your observation. It has been amended in line 31 in the revised manuscript.

- Lines 51 to 61: This paragraph is an enumeration of what other authors did in the past. Try to improve the writing, linking some sentences with others, to engage the reader.

This paragraph has been rewritten in the revised manuscript (lines 55-58) following your suggestion.

- Line 97: Missing dot between "(ECMWF) It".

The dot has been added in the revised manuscript.

- Line 306: "907 tornadoes" instead of "907 tornado".

It has been amended as recommended in the line 284 in the revised manuscript.

Dear Reviewer 2,

We would like to thank you for your time and effort in our manuscript. We think that your comments have been very useful in improving the paper. Please, find below our replies to each one.

Introduction:

- The focus of the paper is on the comparison between hail and non-hail supercells, so you should remain focused on that also in the introduction. Also, the studies you mention in the introduction are almost exclusively focused on Spain, while you should extend your comparison of hail vs non-hail supercells to the whole Mediterranean and possibly other regions.

Thank you for your observation. Unfortunately, there are not much research focused on the differences between hail and non-hail supercells. Hence the importance and the interest of this study. However, we have added some bibliography related to hail thunderstorms in different European regions in the revised manuscript (lines 66-68). *Manzato (2012) recorded hailstorms using hailpads to perform a hail climatology in northeast Italy. Merino et al. (2013) using hailpads data too to study the synoptic and mesoscale configurations for hailstorms in southwestern Europe.*

- Line 45: "supercells in Europe tend to be smaller, both horizontally and vertically, than those formed in the US": do the papers you refer to include quantitative estimation of the reduction in horizontal and vertical extent? I think this is a difficult task to assess, so I would be curious if there are some statistics supporting this statement. Similar considerations apply to the "reduced rotation and shorter life spans": is there any statistics to support your sentence?

It is true that this statement is not supported by quantitative statistics. However, Taszarek et al. (2020b) performed a comparative study for severe convective storms between Europe and US. These authors suggested that severe thunderstorms environments over US are characterized by higher moisture, CAPE, CIN and wind shear than Europe. Based on the convective variables calculated in their study, these authors asserted that thunderstorms over Europe have a lower potential for producing severe weather than those in US. The sentence has been rewritten in the revised manuscripts (lines 45-47):
*Mainly due to orography, smaller Convective Available Potential Energy (CAPE) and wind shear (WS), supercells in Europe tend to be less severe and with return periods (hail ≥ 8cm, violent tornadoes) longer than those formed in the US, and therefore show reduced rotation and shorter life spans (Quirantes et al., 2014; Taszarek et al., 2020b).*

Section 2:

- Has the supercell dataset been validated somehow? For example, did you make a comparison with the hail occurrences as reported in ESWD or in local datasets?

Following your suggestion, the Spanish Supercell Database used here (Martin et al., 2020) has been cross-matched with the ESWD and SINOBAS (Notification System for Singular Atmospheric Observations by AEMET. The validation results show that 81% and 86% of the SP-HAIL events are included in the ESWD and SINOBAS datasets, respectively. The validated result of this dataset has been included in the revised manuscript (lines 92-96).

*In the current study, only the confirmed supercells are selected. These are then categorised as SP-HAIL and SP-NONHAIL according to the observation or not of hail with diameters larger than 5 cm. It should be noted that in this study the Spanish Supercell Database was cross-matched with the European Severe Weather Database (ESWD) and Notification System for Singular Atmospheric Observations (SINOBAS). The validation results show more than 80% of the SP-HAIL events from the Spanish Supercell Database are included in the ESWD and SINOBAS datasets.*

- Line 83: you mentioned earlier that the medium-high confidence events are detected in radar images but without direct observation; here, you mention that thanks to volunteers, 20.5% of the medium-high confidence supercells were confirmed by two-dimensional radar images. Sorry, but I am confused.

This sentence has been rewritten, and some examples were added for a better understanding in the revised manuscript (lines 84-87).

*The supercell sample used is selected from the Spanish Supercell Database (Martín et al., 2020) for the 2011-2020 period. This dataset is formed by confirmed (i.e., doppler radar images, hail greater than 5 cm reports, tornadoes greater than EF2 or images of the event) and medium-high confidence (detected in non-doppler radar images but without direct observation; see Figure 4 in Martín et al. 2020) supercell events through reports from volunteers and collaborators.*

- Line 143: I think it is interesting that the only parameter changing with time is CIN, possibly as a consequence of the change in the environmental conditions after convection is triggered.
- Line 146-147: I disagree with this point. Once convection is triggered, the environment should be "contaminated" by the vertical redistribution of temperature consequent to the vertical motion, thus the profile at tc would be less representative of the environment conducive to supercell development than that at the earlier stage.
- Line 148-149: I do not agree that the information related to WS06 is more important than that on MUCIN. The fact that MUCIN is different reveals that the environment has substantial differences between tc and t0, i.e. before and after convection is triggered (you wrote that other buoyancy terms can be evaluated, but I do not see which ones you consider here).

Thank you for your comments. We agree that it is possible that once the convection is triggered, the environment should be "contaminated" and consequently the results at $t_c$ would be less representative. Therefore, following your suggestion, the results in the revised manuscript are referred only at $t_0$ and explained at the methodology section. Hence, the sentences related to your comments have been deleted in the revised manuscript.

Section 3:

- Line 164: I do not understand why the eastern half of Spain should be special from the point of view of upper-level forcing for ascent.

The eastern area of Spain plays an important role in the upper-level forcing due to the singularity of a complex orography (with the presence of several mountains systems and the Ebro valley, as it can be seen in Figure 1a) with a warm Mediterranean sea in summertime. Indeed, many researches have selected this area for their studies about convection (Castro et al., 1992; Gayà, 2011; Rodriguez and Bech, 2018; Mora et al., 2019;

Gutierrez and Kumjian, 2021). Moreover, as it can be seen in Figure 2a, most of the supercell events are concentrated in eastern of Spain.

- Line 170: in other Mediterranean areas the peak of hailstorms occurs in June (e.g., Manzato, 2012), due, as expected, to a combination of strong diabatic heating and cold air intrusions, more frequent in late spring-early summer. Why does Spain behave differently?

Thank you for the reference. There are some differences related to hailstorms monthly distribution between Manzato (2012) and the current study. These differences might be due to the strongest insolation (irradiance) in Spain in comparison with the irradiance in northeast Italy (see the Figure below), where Manzato (2012) locate their study. Also, it is noteworthy that the current study is more restrictive, since it only considers hail when it comes from supercells and not from all the hailstorms. Moreover, Merino et al. (2013) studied the monthly hailstorm distribution in Southwestern Europe, producing very similar results (see their Figure 5) to the ones obtained by us.

[Figure]

**Source**: Sancho Ávila, J. M., Riesco Martín, J., Jiménez Alonso, C., Sánchez de Cos, M. D. C., Montero Cadalso, J., & López Bartolomé, M. (2012). Solar Radiation Atlas in Spain using EUMETSAT Climate SAF data. Spanish Meteorological Agency (AEMET). https://repositorio.aemet.es/bitstream/20.500.11765/2531/1/atlasradiacion_cal2013.pdf

- Figures 3, 4, 5: To highlight the differences, I suggest showing the SP-HAIL fields and the differences compared to the SP-NONHAIL fields. In the present version, it is difficult to detect the rather small differences. In addition, in Figure 5: contour lines are very difficult to identify, differences are not clear, coastlines can be hardly identified.

Thank you for your suggestion. The SP-HAIL and SP-NONHAIL differences are showed in the corresponding figures in the revised manuscript. Following your advice, contour lines of the Figures 3, 4 and 5 have been modified for a better identification.

- Line 197: do you mean short or small in amplitude?

Thank you for your observation. Indeed, it is short wave. The line 185 have been modified in the revised manuscript.

- Line 205 and elsewhere: moisture, not moist.

The line 193 has been amended as recommended in the revised manuscript.

- Line 208 and elsewhere: easterly winds, not eastern.

The line 196 has been amended as recommended in the revised manuscript.

- Line 209-210: I would rather say that the difference in DWPT is mainly a consequence of the different dominant seasons in the two supercell datasets.

The Figure 2b shows the supercells monthly distribution, depicting the maximum in July for SP-NONHAIL and August for SP-HAIL events corresponding both maxima to the same season (summertime). Notwithstanding, we have calculated the DWPT climatology using the ERA5 dataset for July and August (Figures below). It can be seen some differences between the two months, being the DWPT higher in August (dominant in SP-HAIL) than July (dominant in SP-NONHAIL) in the study area. As mentioned in the line 197 in the revised manuscript the DWPT differences originate in the advection of humid air from the Mediterranean, which is higher in SP-HAIL than SP-NONHAIL events. This fact can be also observed in the Figures below, being this advection stronger in August than July.

[Figure]

July                                    August

Despite the aim of this manuscript is devoted to the analysis of the differences between SP-HAIL and SP-NONHAIL events and not the supercells monthly distribution, we have added a sentence related to this DWPT climatology in the revised manuscript (lines 196-200).

*This would be a result from the geopotential and thermal low configuration described above, which induces humid air advection from the Mediterranean Sea. According to the DWPT climatology (not shown), the DWPT in the Ebro Valley and the Mediterranean coast is higher in August (when the SP-HAIL are predominant; Figure 2b) than in July. The convective processes are then supported by the favourable environment that promotes deep convection in those zones and pushed by the south-westerly flows.*

Lines 211-213: "The high elevations reduce the role of convective inhibition, which is also met by the convergence of southwestern and eastern surface winds": what do you mean??? do you mean that the orography forces the air parcels to be lifted above the LFC?
We agree that this sentence is confusing. The sentence has been deleted in the revised manuscript.

- Line 223: "This maxima omega area matches with positive Q-vector divergence values … and convergence of Q-vectors": I do not understand: do you mean that maxima omega values are superimposed with both divergence and convergence areas???

We agree that the comparison between Q-vector divergence and omega vertical velocity are confusing. For this reason, we have deleted this sentence and the authors have explained the Figure S1 (Q-vectors) in a clearer form in lines 218-221 in the revised manuscript:

- Line 225: "higher values of maxima omega in SP-NONHAIL at 850-500 hPa": why do you consider in the following analysis only the maxima omega vertical velocity at 700-400 hPa thickness in SP-HAIL and not the maxima omega in SP-NONHAIL at 850-500 hPa?

In the revised manuscript, the omega vertical velocity is selected at the 700-400 hPa layer for both SP-HAIL and SP-NONHAIL. The sentence related to the SP-NONHAIL maxima omega at 850-500 hPa has been deleted to avoid misunderstanding to readers.

- Line 227: wind convergence does not enhance and reinforce convection, rather favors triggering.

Thank you for your appreciation. Lines 211-213 of the revised manuscript include the wind convergence effect to trigger the convection as well as the effect of sustained omega vertical velocities and wind convergence to reinforce the deep-moist convection.

- Line 231-232: I am very confused, I see values of order 30 m/s, never below 20 m/s, in Fig. 5.

The Figure 5 has been redone to show in a better way the 0-6 km wind shear (green lines). Indeed, the minimum WS in the study area is 22 m/s. Therefore, the references to values lower than 20 m/s have been deleted in the revised manuscript.

- Line 245: 90-th percentile with respect to what?

The 90th percentile is respect to the MUCAPE values of the dataset to show the vertical profile of the largest and severe supercells as it is mentioned in line 233 in the revised manuscript.

- Line 247: what do you mean with "a better buoyancy distribution"?

We agree this sentence is confusing by which it has been deleted in the revised manuscript.

- Line 248-249: I do not see where the value of CAPE is reported, and where you show that the CAPE values are larger at t0 than at tc;

Please, see the next reply.

- Line 250: "The CIN in SP-HAIL increases from t0 to tc": where do you show this increase? also, it is very hard to physically understand why CIN increases: should not the convection remove progressively the inhibition?

The Figure 5 has been improved to show the CAPE, CIN, LCL and LFC values for each event.

- Line 252: -137 J/kg is a rather extreme value for CIN, I do not believe that convection can develop even in the presence of mountains with such a value; conversely, a value for about -50 J/Kg as reported by Taszarek et al. (2020b) (Line 317) appears more reasonable.

Due to the changes in the methodology, the new CIN value is -94.2 J/kg. We think that the CIN values obtained by Taszarek et al. (2020b) are lower than herein, as their study is not focused on a complex orography domain. Moreover, the CIN value of the revised manuscript corresponds to supercells with the highest MUCAPE values (90th percentile) of the Spanish Supercell Database.

- Line 254-255: "a higher LCL is related to the width of the deep convective updraft, resulting in a wider, deeper, and faster vertical velocity": I do not understand how general this result is and the physical reasoning for that;

According to Mulholland et al. (2021), higher LCLs produce wider and stronger convective updrafts than environments with comparatively lower LCL. Rising dry thermals in simulations with higher LCLs had more time to entrain conditionally unstable air and expand in size before they reached the LCL, setting the stage for wider moist updrafts above the LCL (see Figure 5 of Mulholland et al., 2021).

[Figure]

**Figure 5.** Schematic depicting the processes driving the relationship between lifting condensation level (LCL) height and convective updraft width (denoted by R, or cloud radius). The yellow-orange-red "bubble" near the surface denotes a warm and dry surface thermal that is ascending. Black dashed arrows represent the vertical air motion of the ascending warm and dry surface parcel. Dashed black vertical lines denote the expansion of the ascending dry thermal owing to entrainment, which is represented as black straight and half-circle solid arrows. The purple horizontal lines denote the LCL, and clouds are shown in blue. Green arrows represent the vertical air motion within the cloud and the arrow width denotes the relative magnitude. Acronyms are defined in the text.

Nevertheless, a more detailed explanation has been included in the revised manuscript (lines 239-242):

*According to Mulholland et al. (2021), a higher LCL is related to the deep convective updraft width. This is resulting on a wider and deeper column and a faster vertical velocity due to the larger distance and residence time of the dry thermal to entrain. Thus, the ensuing moist updraft above the LCL is wider, less dilute and has a greater vertical velocity, which would be in line with the omega vertical velocity results (Figure 5b).*

- Line 257: why is it relevant to have large WS values "above" the updraft height?

This sentence has been clarified in the revised manuscript (lines 243-245).

*According to Markowski and Richardson (2010) WS tends to enhance the organization, severity, and longevity of the deep moist convection. This due to the degree to which precipitation and outflow affect with an updraft is reduced as the WS over the updraft depth increases.*

- Line 260: "The evolution from t0 to tc depicts a reduction in WS for SPHAIL": where do you show this point?
- Line 261: what does "contrary to the SP-NONHAIL episodes" refer to?

In the revised manuscript only the sounding at $t_0$ is depicted in Figure 6. Therefore, these sentences have been deleted.

- Line 266: "Different distributions can be seen in Figure 7": I would say this is not relevant, it is rather a consequence of the different dominant seasons in the two categories.

In the revised manuscript, the most remarkable differences in the violin plots are exposed. In this sense, the explanation of the two statistically significant different variables, T2M and DWPT in Figure 7 has been shown, describing differences between SP-NONHAIL and SP-HAIL variability and median values, and exposing the possible causes that promote these differences (lines 251-254 of the revised manuscript).

*The T2M for SP-HAIL distribution depicts a lower variability and larger median value (Table 3) than the corresponding SP-NONHAIL. The T2M maximum (minimum) for SP-HAIL is 33.0 ºC (16.8 ºC), showing both groups a very similar maximum value, while the minimum is significantly lower (7.9 ºC) for SP-NONHAIL. The DWPT median value for SP-HAIL is greater than for SP-NONHAIL (Table 3).*

- Line 266: what do you mean with "bimodal distribution"? I do not see it in Figure 7.

The T2M distributions have changed in the revised manuscript and no longer follows a bimodal distribution (2 mode values).

- Lines 272-276: I think the differences in humidity are mainly due to the different seasons prevailing in the two categories and not to the different wind features;

The explanation of this comment is similar to the above comment related to the Line 209-210 (Figure 4).

- Line 284: what do you mean "with a lower amplitude"?

Thank you for your comment. It is not amplitude, but range. This mistake has been amended in the revised manuscript.

- Lines 287-288: I would rather state that MLCAPE is very close to SBCAPE.

Thank you for your suggestion. This suggestion has been included in the revised manuscript (line 271).

- Line 304: you cannot compare values in high-resolution models with those in reanalyses.

We agree and this sentence has been deleted in the revised manuscript.

- Line 305-307: "the CAPE values found in our study would correspond with those for tornadic storms … finding SBCAPE values higher than 400 J kg-1 in tornadic storms": here you find that 75% are below 400 J/kg, so they do not correspond.

Thank you for your observation. These results have changed in the revised manuscript since the methodology considers only the supercell cycle life results at $t_0$. Therefore, this sentence has been modified indicating that the SBCAPE values from Rodriguez and Bech (2018) are similar to the SBCAPE values for SP-HAIL events in our study (lines 282-283).

- Figures 8, 9, 10: what time do the figures refer to?

In the revised manuscript all figures are referred to $t_0$.

- Lines 319-324: the causes you address for the high CIN would be relevant in case you consider soundings at times distant from the development of the cell, while it is very strange that you have such a high CIN during or in the proximity of convection.

Due to the CIN results changed, this sentence has been deleted in the revised manuscript.

- Line 336 "the median MLLCL in SP-HAIL events is greater": this may be due to the presence of the mountains.

The MLLCL results have changed in the revised manuscript. The MLLCC median value now matches with the Púčik et al. (2015) study.

- Line 337: how do you interpret physically the higher LFC for SP-HAIL? I would rather expect that a lower LFC would favor deeper updrafts and then stronger hail formation!
- Line 339: values of MLLFC higher than 2000-3000 m appear extremely high (you even obtain values of 5000 J/kg!): how do you explain them?

The MLLFC results have changed in the revised manuscript, resulting in not statistically significant MLLFC differences between SP-HAIL and SP-NONHAIL.

Related to your next question, the MLLFC values in the revised manuscript have diminished, with the centred 50% of the MLLFCs between 1500 and 2400 m with the 90-percentile around 3500 m.

The next figure displays a supercell sounding corresponding to a particular supercell of the used data base in the nearest ERA5 grid point. As it can be seen the MLLFC is 3070 m. It is worth to note that the original method to compute the different convective variables was based on 9 grid points around the centre of each supercell. However, and following the reviewer's 3 suggestion, the methodology has changed, and all the convective variables are analysed in the nearest grid point at $t_0$ in the revised manuscript.

[Figure]

Moreover, as an example of high MLLFC values, the Spanish Meteorological Service reported on 2022-06-29 values of 3521.8 m (NCL value in the sounding figure) in Teruel, located at ~ 40ºN, -1ºW, with higher reflectivity values (right figure of the panel) observed by radar in the domain.

[Figure]

- Lines 350-355: Figure 10 shows the opposite compared to what you wrote, i.e. wind shear is higher for SP-NONHAIL.

It was a mistake, and it has been amended in the revised manuscript.

- Line 363: about SHR01 "the median value of SP-NONHAIL is higher than SP-HAIL": this appears counterintuitive: any explanation for that?

The median value of SP-HAIL is higher than SP-NOHAIL in the new results of the revised manuscript.

Dear Reviewer 3,

We would like to thank you for your time and effort in our manuscript. We really appreciate the detail of your reading, as your comments have been very useful in improving the paper. We have addressed all your questions. Please, find below our replies to each one.

Major comments:

I also have a feeling that analysis of differences between t0 and tc is a redundant part of this study as it doesn't introduce important findings. This is especially strange given that authors try to find differences in small details over small distances between t0 and tc, but at the same time they average their profiles to 9x9 grids and do not benefit from 0.25 deg resolution of ERA5. Trying to evaluate subtle differences among closely located t0 and tc for large synoptic-scale features at figures 3, 4 and 5 is even less scientifically relevant. While I like the concept of dividing supercells into hail producing and non-producing events, I am just skeptical whether division into t0 and tc is worth all the attention authors devote in this study. This is not a major issue and I leave the decision regarding incusion/exclusion of this part to authors. At the end of the day it is their decision what and how they want to present in their work. However, for future studies with this dataset, instead of ERA5 with 9x9 grid averaging, a convective-allowing high-resolution simulation would be likely more appropriate to evaluate different stages of the supercell lifecycle at t0 and tc and investigate the influence of ambient environment and local orographical features.

Following your suggestion, the analysis of differences between $t_0$ and $t_c$ has been deleted in the revised manuscript. Moreover, we agree, and we have focused the analysis on SP-HAIL and SP-NONHAIL differences.

Following your advice, we have also calculated the convective variables in the nearest grid point of the supercell initial time ($t_0$) in the revised manuscript. Therefore, the 9x9 grid has been deleted and the cycle life analysis has been changed ($t_c$ to $t_0$).

In this work, I think that division of results into high-CAPE and high-shear events would be probably more interesting and scientifically important in the context of other similar work that has been done for Europe (compared to t0 and tc approach). It is well known that European severe storms are mostly driven by strong kinematics and in lower degree by high instability, which is also a case for supercells.

Thank you for your suggestion. We have selected the high-CAPE (90th percentile) events to depict the composite soundings (Figure 6) since CAPE differences are statistically significant between both supercell groups, while there are not differences for the high-wind shear events (>20m/s).

Minor comments:
- L15: Suggest changing to „the synoptic configurations and proximity atmospheric profiles related to the supercell events".

It has been amended as recommended.

- L19-L21: Awkward sentence construction, please rewrite for clarity. Perhaps splitting this sentence into two can help.

Thank you for your observation. This sentence has been clarified in the revised manuscript (lines 19-21).

- L27: Suggest changing „life" to „lifecycle"

It has been amended as recommended.

- L35: I am not entirely sure I can agree with this sentence and the phrase „easily detected". Supercell detection in Europe is generally not easy if high-quality Doppler radar data is not available (like in the U.S.). I am also not sure how a mesocyclone (which is a core definition of the supercell thunderstorm) can be detected by lightning data. In the majority of instances we can only suspect that supercell thunderstorms developed based on its morphological features, but only a small fraction of these events can be captured by nearby Doppler radar velocity products that provide ultimate confirmation of the mesocyclone. I suggest authors reword and soften this sentence or remove it.

Following your recommendation, this sentence has been modified (*easily detected* has been removed) in the revised manuscript (line 35).

- L43: No need to use „observational" ahead of „reports". Authors may consider using „severe weather reports" instead.

It has been amended as recommended.

- L45: Is there any scientific proof that they are indeed smaller? Authors speculate that it is due to oroghraphy and land-sea interactions, but is it really the case? What about big supercells in Nebraska or Southeastern U.S. along the coast of GOM? Is there any scientific proof showing that orography acts to reduce size of the supercells? Perhaps weaker supercells in Spain are rather due to smaller CAPE and WS / less favorable wind profile compared to their U.S. equivalents. I suggest rewording.

Taszarek et al. (2020b) performed a comparative study for severe convective storms between Europe and US, suggesting that severe thunderstorm environments over the US are characterized by higher moisture, CAPE, CIN, wind shear than Europe. Based on the convective variables calculated in their study, these authors asserted that thunderstorms over Europe have a lower potential for producing severe weather than those in US. Nevertheless, the sentence has been modified in the revised manuscript (lines 45-47).
*Mainly due to orography, smaller Convective Available Potential Energy (CAPE) and wind shear (WS), supercells in Europe tend to be less severe and with return periods (hail ≥ 8cm, violent tornadoes) longer than those formed in the US, and therefore show reduced rotation and shorter life spans (Quirantes et al., 2014; Taszarek et al., 2020b).*

- L59: Which „other regions of the world"? Please be more specific.

Thank you for your observation. This sentence has been rewritten in the revised manuscript (lines 55-57).
*Tornadoes occurrence and intensity are not as severe as in other regions of the world (e.g., US) mainly due to the absence of wet fluxes inland (Rodríguez and Bech, 2018).*

- L95-101: Did authors also use surface data in addition to pressure levels, and eliminate all pressure levels falling below orography for the purposes of parameter calculations? This information should be included in this paragraph. Also, which software was used to calculate convective parameters. SHARpy, MetPy, other, or your own scripts? Did you also consider that some of the proximity profiles may be contaminated by the convection ongoing in ERA5? Did you use convective

precipitation threshold equalling 0mm to eliminate such profiles? This might be an approach worth considering in potential future studies to make sure evaluated profiles are pre-convective.

We agree that probably the proximity profiles may be "contaminated" once the convection is triggered. Therefore, as abovementioned, we have recalculated all the parameters using the ERA5 hybrid model levels and the supercell grid point nearest to $t_0$. The software used to calculate the convective variables is thundeR (rawinsonde package). Some of this information has been included in the methodology Section of the revised manuscript. Related to the used of convective precipitation threshold, thank you for your suggestion and we note for future supercell research.

- 108-111: I am not sure if that was a good idea. In this way authors do not benefit from the superior (compared to other reanalyses) resolution of ERA5. This averaging can have an impact on areas with complex orography and result in the loss of important details. Did authors try to reproduce their results without a 9x9 grid averaging approach? Were these results much different?

Following your suggestion, the convective variables have been calculated in the nearest grid point to the supercell at $t_0$ instead of 9x9 grid averaged in the revised manuscript.

- L124: What authors mean by „The 2-meter temperature (T2M) and dew-point (DWPT) are computed". In which aspect T2M and DWPT required computations? To avoid using a word „computed" authors can reword into something like „We selected the 2-meter temperature (…)".

We agree and this sentence has been rewritten in the revised manuscript (line 125).

- L126: What depth was used for calculating mixed-layer?

Thank you for your observation. The mixed-layer is averaged over 0-500 m above ground level. This information has been added in the revised manuscript (line 126).

- L149: I am not sure if I understand what authors mean by „This variable is much more interesting than MUCIN, as there are other buoyancy terms which can be evaluated"

Thank you for your comments. Following your suggestion, the results in the revised manuscript are referred only at $t_0$. Hence, the sentence related to your comment has been deleted in the revised manuscript.

- L205: Change „moist" to „moisture".

It has been modified as recommended (line 193 in the revised manuscript).

- L244: Why do authors think that 90th percentile of MU_CAPE would indicate „largest and severe supercells"? Suggest rewording to „of the supercells developing in highly unstable environments". Instead of providing mean skew-t profiles divided into t0 and tc it could be potentially interesting to provide also mean skew-t profiles for 90th percentile of WS events as high WS is a major contributor to severe storms in Europe compared to instability that is often limited.

We agree and this sentence has been modified as suggested (line XX in the revised manuscript).

Related to the second comment: It would be interesting to provide mean skew-t profiles for 90th percentile of WS; however, the WS does not present statistically significant differences between SP-HAIL and SP-NONHAIL events. For this reason, we only

presented skew-T profiles for the 90th percentile of MUCAPE (with statistically significant differences).

- L256: What exactly „helps to organize convection"? Please rewrite for clarity.

It has been rewritten in a clarified form in the revised manuscript (lines 242-245).
*Wind barbs reveal a moderate WS06 for both types of supercells. According to Markowski and Richardson (2010) WS tends to enhance the organization, severity, and longevity of the deep moist convection. This due to the degree to which precipitation and outflow affect with an updraft is reduced as the WS over the updraft depth increases.*

- L259: „Also, the sounding composites show large wind values in upper-levels (< 400 hPa), which may favor wind divergence at the upper troposphere and deep-moist convection" – wind values from single profile cannot be used to determine upper-tropospheric divergence and deep-moist convection. It is a spatial pattern of the pressure field that allows to determine divergence and potential areas for the large-scale lift that may trigger deep moist convection. Please rewrite.

Thank you for your observation. You are right and this sentence has been deleted in the revised manuscript.

- L260: „The evolution from t0 to tc depicts a reduction in WS for SP- HAIL, which is mainly denoted in the wind speed and not in the rotation" – I do not understand what authors mean by „and not in the rotation". The degree of veering in the vertical wind profile?

As the evolution from $t_0$ to $t_c$ is not represented in the revised manuscript, this sentence has been deleted.

- L272: „These differences are mainly originated in the low-level wind flows." - awkward sentence construction, please rewrite for clarity.

It has been rewritten in a clarified form in the revised manuscript (lines 254-256).
*These differences are mainly originated in the wind flows, since in the Spanish Mediterranean area, Balearic Islands and places favourable for maritime fluxes, the main contributor to low-level moisture is advection from the warm Mediterranean Sea.*

- L280: CAPE can be a useful predictor but only with the combination of vertical wind shear. Over the tropics there is plenty of CAPE but rarely any supercell or large hail due to weak WS.

Thank you for the explanation. It has been added in the revised manuscript (line 262).

- L286: How CAPE can be dependent on the orography? Please be more specific. Over northern Great Plains CAPE can reach as high as 9000 J/kg over higher elevation in Nebraska.

According to Weisman and Klemp (1982) and Markowski and Dotzek (2011) CAPE is slightly larger in high altitudes than flat terrain because potential temperature increases evenly with height.
This explanation has been added in the revised manuscript (lines 268-271).
*According to Weisman and Klemp (1982) and Markowski and Dotzek (2011), CAPE is dependent on humidity and orography, with slightly larger values in high elevations than in low terrains because potential temperature increases evenly with height. Therefore, the differences between the current study and Kaltenböck et al. (2019) lie in the high elevations and relatively low humidity in the research area.*

L288: Larger compared to what?
This sentence was wrong, and it has been modified in the revised manuscript (lines 271-272).
*The MLCAPE median value is close to the SBCAPE value and both also yield larger results for SP-HAIL than for SP-NONHAIL events.*

- L311-L312: I believe this sentence is inaccurate. It is not only an ERA5-related issue but nearly every reanalysis (or NWP dataset) and is related to limited vertical resolution of available levels. applied convective parameterizations and convective contamination. Given that authors used less numerous pressure levels (instead of more frequent sigma levels), CIN values are expected to be less accurate as well. However, as shown in other studies, compared to other reanalyses ERA5 still performs better for CIN (e.g. table 2 in https://doi.org/10.1175/JCLI-D-20-0484.1). I suggest to soften this sentence and reword it to something like: „It is well known that due to limited vertical resolution reanalyses do not represent capping inversions very well".

Thank you very much for your explanation. It has been amended in the revised manuscript (lines 292-293).
*It is well known that due to limited vertical resolution reanalyses do not represent capping inversions very well (Nevius and Evans, 2018; Coffer et al., 2020; Taszarek et al., 2021).*

- L321 Airmass advections from NW Africa and development of elevated mixed-layers can be also another reason for higher CIN across Spain and W part of Mediterranean compared to other parts of Europe.

Due to the change of the CIN results, this sentence has been deleted in the revised manuscript.

- L325: „a mechanical trigger to force the mechanism that initiates convection,, – awkward sentence construction, please rewrite for clarity.

It has been rewritten in a clearer form in the revised manuscript (lines 302-304).
*Therefore, a mechanical trigger (e.g., air parcels lifted by orography or low-level convergence wind) is required to force initiation of convection to overcome the LCL. The conjunction of these factors favours great vertical motions and organized convection.*

- L357: Helicity or rather storm-relative helicity?

You are right. It has been amended in the revised manuscript in line 336.

- L359: Period missing before „Environments". Also, this sentence has an awkward construction, please rewrite for clarity.

This sentence has been rewritten in a clearer form (lines 336-339) in the revised manuscript. *Storm-relative helicity (SRH) is a frequent parameter used for forecasting supercells and tornadoes since it quantifies the cyclonic updraft rotation in right and left moving supercells (in this survey only the right-moving measure is used; Davies-Jones et al., 1990; Bunkers et al., 2002). Higher SRH values are usually related to the development of the mesocyclones and large hail formation (Rasmussen and Blanchard, 1998; Thompson et al., 2003).*

- L387: „Omega vertical velocity reveals that the SP-HAIL's updraft is higher" – ERA5 omega vertical velocity derived from 0.25 deg grid and averaged by authors

to 9x9 matrix surely does not tell anything about local storm-scale convective updraft.

We agree and this sentence has been deleted in the revised manuscript.

- Figure 3, 4 and 5: Text that is at the top of each figure and x and y axis is too small and impossible to read.

It has been amended as recommended in the revised manuscript.

- Caption to figure 6. 90th percentile of what? Please be more specific in the figure caption.

It is referred to the MUCAPE values. It has been added in the figure caption in the revised manuscript.

---

## Author Response (AR2)

Carlos Calvo Sancho. *e-mail*: carlos.calvo.sancho@uva.es
Department of Applied Mathematics.
Faculty of Computer Engineering. University of Valladolid.
Segovia, Spain.

WEATHER AND CLIMATE DYNAMICS

Manuscript number wcd-2022-27

Segovia, August 14th, 2022

Dear Editor:

On behalf of all the co-authors, I am pleased to submit the second revised version of the research article entitled "Supercell Convective Environments in Spain based on ERA5: Hail and Non-Hail Differences" by Calvo-Sancho et al. for consideration in the journal Weather and Climate Dynamics.

First of all, we thank the reviewers for their time and constructive comments. We believe they have significantly helped to improve the quality of the paper. We hope that the corrections we have made will be in accordance with the "Weather and Climate Dynamics" standards and expectations. Please find attached the detailed answers to all the comments and queries provided by the reviewers.

Thank you very much.

Yours sincerely,

Carlos Calvo Sancho

Dear Reviewer 2,

We would like to thank you for your time and effort in our manuscript. We really appreciate the detail of your reading, as your comments have been very useful in improving the paper. We have tried to address all your questions. Please, find below our replies to each one.

General: manuscript needs language proofreading from a native speaker because it contains errors and does not meet publishing quality.
We have tried to detect and amend the linguistic errors to our best knowledge.

L35 A reference to Blair et al. (2017) may be also relevant in this context.
Thank you for your suggestion. This reference has been added in the revised manuscript.

L35: "Supercells are common phenomena in spring and summer (Brooks et al., 2019), and can be detected through ground-based or satellite lightning detection systems" - I still think that supercells cannot be detected through satellite or lightning detection systems. While both of these systems can be useful as supporting information, Doppler radar data is a primary source to confirm a mesocyclone. Please rewrite.
Thank you for your comment. We agree and this sentence has been rewritten in the revised manuscript (lines 35-38).
*Supercells are common phenomena in spring and summer (Brooks et al., 2019), and can be detected through Doppler radar data to confirm the associated mesocyclone (Blair et al., 2011; Kahraman et al., 2017). Moreover, ground-based or satellite lightning detection systems can be useful as supporting information (Bedka et al., 2018; Galanaki et al., 2018).*

L46: I am not sure if referenced literature says anything about "reduced rotation and shorter life spans" of the supercells in Europe compared to their U.S. counterparts.
According to Quirantes et al. (2014) mesocyclones in Spain tend to be smaller, horizontally and vertically, in general, than the classic ones in the U.S., and therefore present lower rotation velocities and shorter life spans. Nevertheless, an important severity is still present in any type of supercell. Consequently, in Spain the environmental parameters related to severe convective weather are generally lower than in North America and, therefore, the mesocyclone rotations are both less intense and with shorter life spans.
https://repositorio.aemet.es/bitstream/20.500.11765/709/10/Caracteristicas_supercelulas.pdf

L85: "Doppler" with capital letter (also throughout the manuscript).
Thank you for your observation. It has been amended in the revised manuscript.

L98: "may not represent conditions in which supercell developed"
It has been amended in the revised manuscript following your suggestion.

L127: Authors should use an en-dash symbol when describing intervals, e.g. "0–6 km" instead of "0-6 km".
Thank you for your observation. The en-dash symbol has been used in the revised manuscript.

L207: "(referred to absolute values)" - I am not sure what "absolute values" mean in this context.
We agree and this explanation has been deleted in the revised manuscript.

Figure 5: Wind shear isoline step of 1 m/s is probably too detailed. A step of 2, 4 or even 5 m/s would improve readability of this figure.
Thank you for your recommendation. This figure has been redone depicting less wind shear isolines.

L242: Updraft speed of a local-scale convective cell has likely nothing to do with large-scale ascent denoted by ERA5 coarse grid (omega denote vertical changes of around 0.2 Pa/s on Figure 5). Please rewrite.
This sentence has been deleted in the revised manuscript.

L244: "This due to the degree to which precipitation and outflow affect with an updraft is reduced as the WS over the updraft depth increases." - it is hard to understand what the authors mean here.
We agree and finally this sentence has been deleted in the revised manuscript.

L296: I am skeptical whether we can write about capping inversion layers when the median of MLCIN is -20 J/kg. Perhaps rather the lack of inversion layers or very weak inhibition?
Indeed, you are right, and this sentence has been amended in the revised manuscript (line 301).
*The SP-HAIL events show a very weak inhibition (Figure 8b) with a MLCIN median of -20.2 J kg-1 (Table 3), and of -12.2 J kg-1 for SP-NONHAIL*

L302: What is a "mechanical trigger"?
In order to clarify, we have used "mechanical lifting" instead of "mechanical trigger" in the revised manuscript (line 307).

L302: "Therefore, a mechanical trigger (e.g., air parcels lifted by orography or low-level convergence wind) is required to force initiation of convection to overcome the LCL" - Why LCL? Perhaps the authors confused it with LFC. A layer between LCL and LFC can still contain stable layers that will prevent storm development.
Indeed, it was a mistake, and this sentence is referred to LFC. It has been amended in the revised manuscript (line 309).

L336: I don't think that SRH "quantifies the cyclonic updraft rotation". It is rather a metric of environmental helicity that can support development of rotating updrafts.
We agree and this SRH definition has been amended in the revised manuscript (lines 343-344) following your recommendation.
*Storm-relative helicity (SRH) is a frequent parameter used for forecasting supercells and tornadoes since it quantifies the streamwise vorticity that can support the development of rotating updrafts in right and left moving supercells the cyclonic updraft rotation in right and left moving supercells (in this survey only the right-moving measure is used; Davies-Jones et al., 1990; Bunkers et al., 2002).*